# Lorentz-violating type-II Dirac fermions in transition metal dichalcogenide PtTe$_2$

Mingzhe Yan[1], Huaqing Huang[1], Kenan Zhang[1], Eryin Wang[1], Wei Yao [1], Ke Deng[1], Guoliang Wan[1], Hongyun Zhang[1], Masashi Arita[2], Haitao Yang[1,3], Zhe Sun[4], Hong Yao[5,6], Yang Wu[3], Shoushan Fan[1,3,6], Wenhui Duan [1,6] & Shuyun Zhou[1,6]

Topological semimetals have recently attracted extensive research interests as host materials to condensed matter physics counterparts of Dirac and Weyl fermions originally proposed in high energy physics. Although Lorentz invariance is required in high energy physics, it is not necessarily obeyed in condensed matter physics, and thus Lorentz-violating type-II Weyl/Dirac fermions could be realized in topological semimetals. The recent realization of type-II Weyl fermions raises the question whether their spin-degenerate counterpart—type-II Dirac fermions—can be experimentally realized too. Here, we report the experimental evidence of type-II Dirac fermions in bulk stoichiometric PtTe$_2$ single crystal. Angle-resolved photoemission spectroscopy measurements and first-principles calculations reveal a pair of strongly tilted Dirac cones along the $\Gamma$-A direction, confirming PtTe$_2$ as a type-II Dirac semimetal. Our results provide opportunities for investigating novel quantum phenomena (e.g., anisotropic magneto-transport) and topological phase transition.

[1] State Key Laboratory of Low Dimensional Quantum Physics and Department of Physics, Tsinghua University, Beijing 100084, China. [2] Hiroshima Synchrotron Radiation Center, Hiroshima University, Higashihiroshima, Hiroshima 739-0046, Japan. [3] Department of Physics and Tsinghua-Foxconn Nanotechnology Research Center, Tsinghua University, Beijing 100084, China. [4] National Synchrotron Radiation Laboratory, University of Science and Technology of China, Hefei, Anhui 230029, China. [5] Institute of Advanced Study, Tsinghua University, Beijing 100084, China. [6] Collaborative Innovation Center of Quantum Matter, Beijing, China. Mingzhe Yan, Huaqing Huang and Kenan Zhang contributed equally to this work. Correspondence and requests for materials should be addressed to Y.W. (email: wuyang.thu@gmail.com) or to W.D. (email: dwh@phys.tsinghua.edu.cn) or to S.Z. (email: syzhou@mail.tsinghua.edu.cn)

In three dimensional topological semimetals, the low energy excitations—Dirac or Weyl fermions—are described by the massless Dirac equation[1]. Dirac fermions are protected by certain crystal symmetries[2–4]. When time-reversal or inversion symmetry is broken, a spin-degenerate Dirac fermion splits into two Weyl fermions, and the topological surface states (TSS) evolve from a closed Fermi surface to open Fermi arcs[5, 6]. The generic Hamiltonian for Dirac and Weyl fermions is $H(\mathbf{k}) = T(\mathbf{k}) \pm U(\mathbf{k})$, where $U(\mathbf{k})$ is a potential component and $T(\mathbf{k})$ is a linear kinetic term that tilts the Dirac cone. The relative magnitude of $T(\mathbf{k})$ and $U(\mathbf{k})$ can be used to classify the topological nature of the Dirac or Weyl semimetals[7]. For type-I Dirac[8, 9] and Weyl semimetals[6, 10, 11], which obey the Lorentz invariance, $T(\mathbf{k}) < U(\mathbf{k})$ and isolated Dirac or Weyl points with linear Dirac cones are expected at the Fermi energy (see schematic drawing in Fig. 1a). When $T(\mathbf{k}) > U(\mathbf{k})$ along certain momentum direction, the Lorentz invariance is violated and strongly titled Dirac cones emerge at the topologically protected touching points between electron and hole pockets (Fig. 1b). Such Lorentz-violating Dirac fermions are classified as type-II Dirac fermions. The classification and comparison of type-I and type-II Dirac/Weyl semimetals are summarized in Table 1.

The different band topology can also lead to distinct magneto-transport properties. While type-I Dirac and Weyl semimetals exhibit negative magneto-resistance along all directions[16–18], the magneto-transport properties of type-II semimetals are expected to be extremely anisotropic and negative magneto-resistance is expected only along momentum directions with $T(\mathbf{k}) < U(\mathbf{k})$[7, 19]. Although type-II Weyl semimetals have been reported recently[14, 15, 20–24], type-II Dirac semimetals still remain to be realized experimentally. This is particularly important since type-II Dirac semimetal stands at the critical point of topological phase transition and can be tuned to Weyl semimetal or topological crystalline insulator via crystal distortions or magnetic doping.

Here we fill in the last missing element in the classification of Dirac and Weyl semimetals. By combining angle-resolved photoemission spectroscopy (ARPES) measurements and first-principles calculations, we report the experimental evidence of type-II Dirac fermions in bulk PtTe2 single crystal with a pair of strongly tilted Dirac cones.

## Results

**Structural characterization.** PtTe2 crystallizes in the CdI2-type trigonal (1 T) structure with $P\bar{3}m1$ space group (No. 162). The crystal structure is composed of edge-shared PtTe6 octahedra with PtTe2 layers tiling the ab plane (Fig. 1c, d). The isostructural PtSe2 film with one monolayer thickness is a semiconductor with a 1.2 eV gap[25] and exhibits helical spin texture with spin-layer locking[26]. Here we focus on the topological property of the bulk semimetallic PtTe2 crystal, and we note that similar topological property is also expected in bulk PtSe2[27]. Figure 1e shows the hexagonal bulk Brillouin zone (BZ) and projected surface BZ onto the (001) surface. The Raman spectrum in Fig. 1f shows the $E_g$ and $A_{1g}$ vibrational modes at ~ 110 and 157 cm$^{-1}$ respectively, which are typical for 1 T structure[28]. The sharp X-ray diffraction (XRD) peaks (Fig. 1g) and low-energy electron diffraction (LEED) pattern (Fig. 1h) confirm the high quality of the single crystals.

**Electronic structure.** The overview of PtTe2 band structure measured by ARPES near the Fermi energy ($E_F$) is shown in Fig. 2. Figures 2a, b show ARPES data taken along two high-symmetry directions Γ-M and Γ-K at photon energy of 22 eV. There are a few conical dispersions in the ARPES data. The most obvious one is centered at the Γ point, which is formed by a V-shaped dispersion touching a flatter Λ-shaped dispersion (pointed by red arrow). The calculated projected spectral weight along the two high-symmetric directions is shown in Figs. 2c, d for comparison. The cone-like dispersion discussed above corresponds to continuous states in the calculation, suggesting that this band is from the bulk states. This cone shows up as a pocket around the Γ point in the measured and calculated Fermi surface maps (Figs. 2e, f). It is also clearly revealed in the

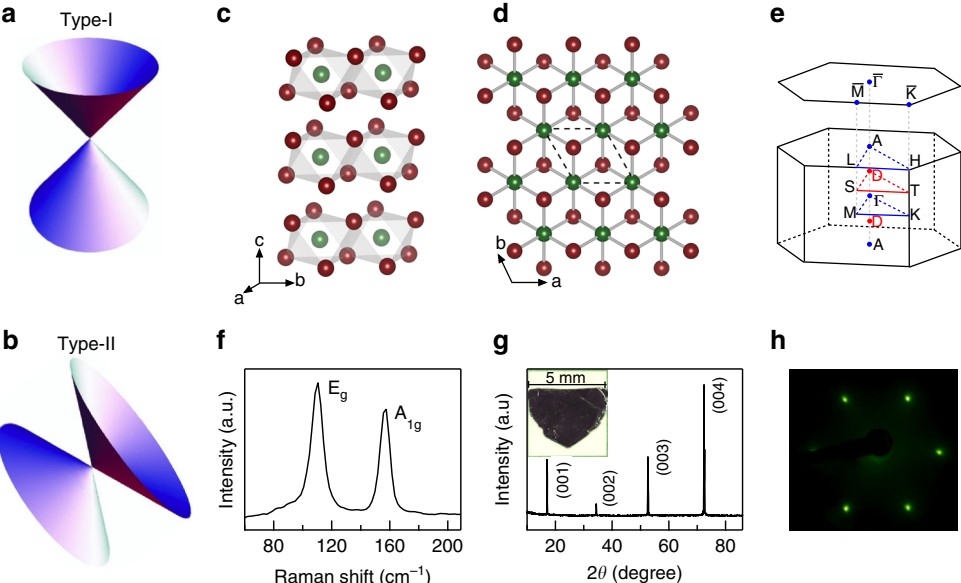

**Fig. 1** Characterization of type-II Dirac semimetal PtTe2. **a**, **b** Schematic drawing of type-I and type-II Dirac fermions. **c**, **d** *Side* and *top* views of PtTe2 crystal structure. *Green balls* are Pt atoms and *red balls* are Te atoms. The unit cell is indicated by *black dashed line*. **e** Bulk and projected surface Brillouin zone onto (001) plane. *Red dots* (labeled as D) mark the positions of 3D Dirac points. **f** Raman spectrum measured at room temperature. **g** XRD of PtTe2 measured at room temperature. The *inset* shows the picture of one single crystal with a few mm size. **h** LEED pattern taken at beam energy of 70 eV

**Table 1 The classification of type-I and type-II Dirac/Weyl semimetal**

| | Classification | Type I | Type II |
|---|---|---|---|
| | | Lorentz invariant | Breaking Lorentz invariance |
| Dirac | Protected by crystal symmetry Doubly-degenerate Dirac cone Closed surface from TSS | $Na_3Bi$[8, 12] $Cd_3As_2$[9, 13] | $PtTe_2$ (this work) |
| Weyl | Breaking time-reversal or inversion symmetry Non-degenerate Dirac cone Open Fermi arcs from TSS | TaAs family[10, 11] | $MoTe_2$[14, 15] |
| Electronic and transport properties | | Linear Dirac cone | Strongly tilted Dirac cone |
| | | Isolated Dirac/Weyl points with vanishing DOS at $E_F$ Negative magneto-resistance along all direction | Dirac/Weyl points at the touching of electron and hole pockets Anisotropic, negative magneto-resistance along certain direction |

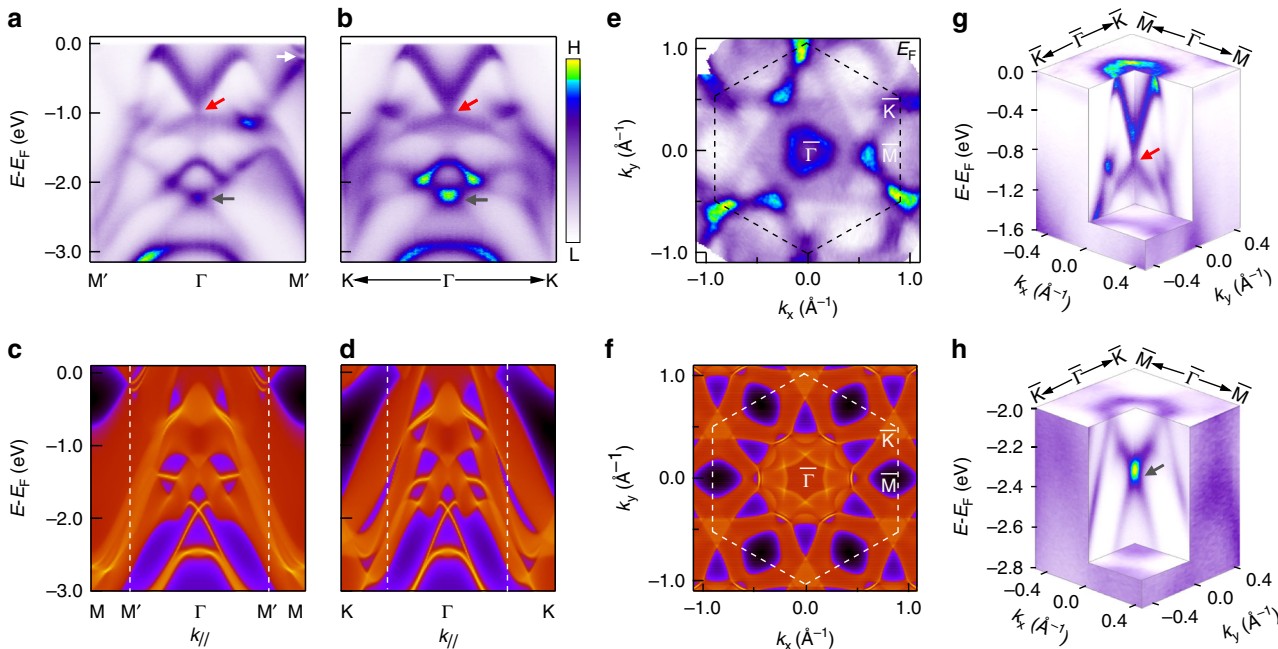

**Fig. 2** The electronic structure of $PtTe_2$. **a**, **b** Band dispersions along the Γ-M (**a**) and Γ-K (**b**) direction at photon energy of 22 eV. *Red* and *gray arrow* indicate the bulk Dirac cone and the surface state Dirac cone respectively. **c**, **d** Calculated band dispersion along the Γ-M (**c**) and Γ-K (**d**) directions using Wannier function. **e** Measured Fermi surface map at photon energy of 21.2 eV. *Black dashed* line indicates surface Brillouin zone. **f** Calculated Fermi surface map. **g**, **h** Three-dimensional E–$k_x$–$k_y$ plots of the bulk Dirac cone around the Γ point **g** and the surface state Dirac cone **h**

three dimensional electronic structure shown in Fig. 2g. The evolution of this cone with the out-of-plane momentum $k_z$ and its topological property are the main focus of this work. The second conical dispersion is located between the Γ and M points (labeled as M′), and it is gapped at the Dirac point slightly below $E_F$ (pointed by *white arrow*). Calculated dispersions (Fig. 2c) show that this cone has bulk properties, and there are sharp surface states connecting the gapped Dirac cone here. The third conical dispersion is at much deeper energy between − 2.0 and − 2.6 eV (pointed by *gray arrow* in Figs. 2a, b). This cone corresponds to sharp surface states in the calculated dispersions (Fig. 2c), and it connects the gapped bulk bands, similar to the $Z_2$ topological surface states observed in $PdTe_2$[29]. The comparison between the measured and calculated band structure shows a good agreement with multiple conical dispersions arising from both the bulk bands and surface states.

To reveal the bulk versus surface properties of these Dirac cones, we show in Fig. 3 ARPES data measured along the Γ-K and Γ-M directions using different photon energies. The corresponding $k_z$ values are calculated using an inner potential of 13 eV[30], which is determined by comparing the experimental data with theoretical calculations. Figure 3a–e shows the measured dispersions. The calculated bulk band dispersions at each $k_z$ value are overplotted on the curvature image in Figs. 3f–j. A good agreement is obtained for the bulk Dirac cone at the Γ point and its evolution with $k_z$. The Dirac point discussed above is at $k_z = 0.35c^*$ ($c^* = 2\pi/c$), which is labeled as D in Fig. 1e. Away from this special point along the Γ-A direction, the valence and conduction bands begin to separate, and the separation becomes larger when $k_z$ moves further away from $0.35c^*$. The strong $k_z$ dependence confirms its three-dimensional nature. We note that at $k_z = 0.30c^*$, some signatures of the dispersion at

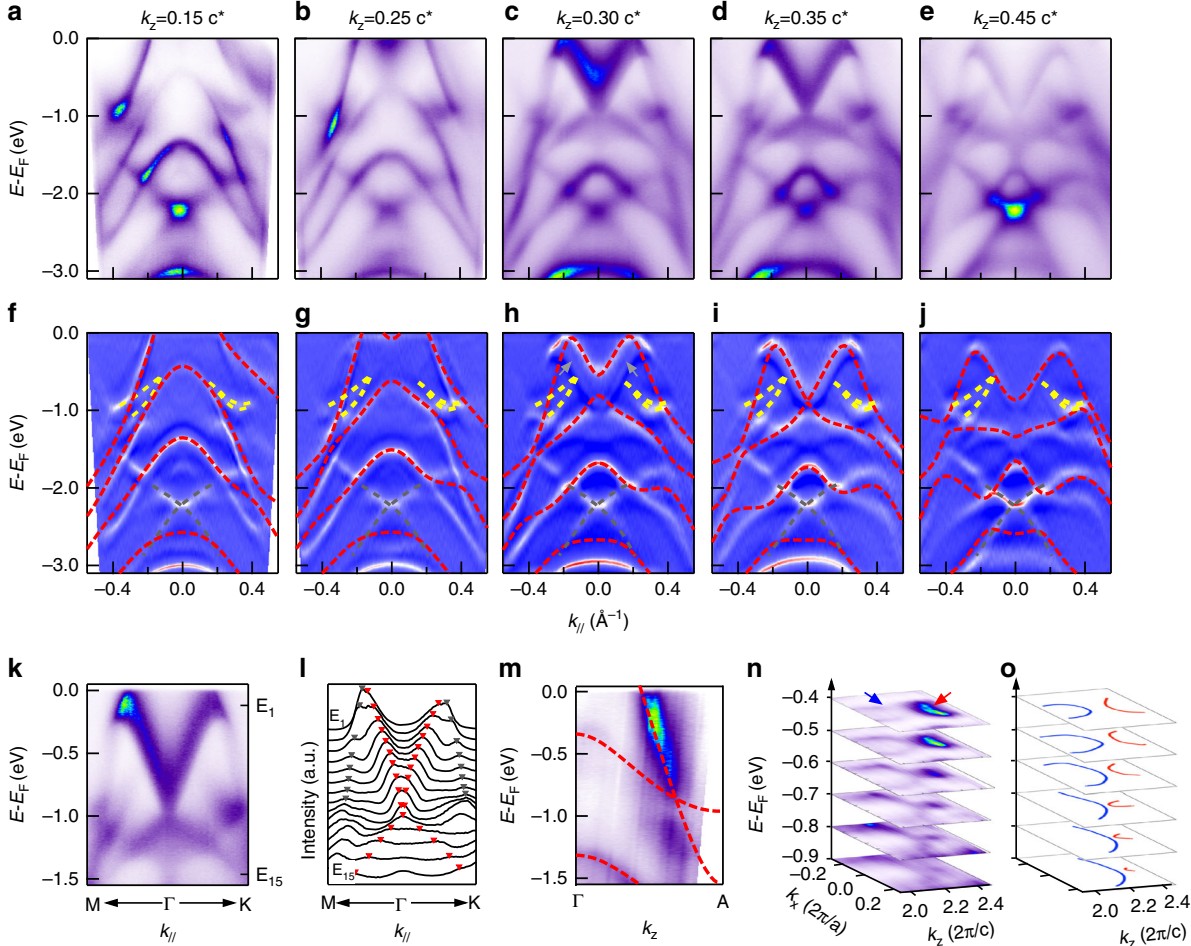

**Fig. 3** Evidence of type-II Dirac fermions in PtTe$_2$. **a–e** Dispersions along the M-Γ-K direction measured at 17, 19, 21, 22, and 24 eV respectively. The corresponding $k_z$ values in the reduced BZ are labeled in each panel. **f–j** Comparison of EDC-curvature with calculated dispersions from first-principles calculations (*red dashed lines*). Surface states are highlighted by *yellow* and *gray dashed line* respectively. **k** In-plane Dirac cone along the M-Γ-K direction measured at 22 eV. **l** MDCs for the data shown in **k**. *Red* and *gray markers* are guides for the peak positions of inner and outer bands. **m** Measured dispersion at $\mathbf{k}_{||} = 0$. *Red broken lines* are calculated dispersions for comparison. **n** Three-dimensional map E-$k_x$-$k_z$ at $k_y = 0$, *blue* and *red arrows* indicate hole and electron pockets. The $k_z$ values are calculated in extended Brillouin Zone. **o** Illustration for the evolution of hole pocket (*blue*) and electron pocket (*red*) with binding energy

$k_z = 0.35c^*$ are also observed (pointed by *gray arrow*), suggesting that there is significant $k_z$ broadening due to a finite escape depth of photoelectrons[31]. In addition to the bulk bands discussed above, there are surface states between −0.5 to −1 eV at the BZ center (highlighted by *yellow dashed line* in Figs. 3f–j) and at deeper energy (*gray dashed line*) that do not change with photon energy.

Figure 3k shows the zoom-in dispersions at the Dirac point. The conical dispersion can be clearly observed by following the peaks in the momentum distribution curves (MDCs) in Fig. 3l. The type-II characteristics are revealed by plotting the dispersion as a function of $k_z$ (Fig. 3m) where a strongly tilted Dirac cone at the D point is revealed. The intensity of the lower branch in the Dirac cone is weaker than the upper one, which is attributed to dipole matrix element effect[32]. We note that a gap-like feature appears at the band crossing point. We have taken a fine $k_z$ step of $0.012c^*$ (corresponding to photon energy step of 0.3 eV, see Supplementary Fig. 1) to exclude misalignment. A more likely possibility is the $k_z$ broadening due to the finite escape depth of photoelectrons. Namely, the measured dispersion is the averaged dispersion over a finite $k_z$ window (from the penetration depth $\lambda \sim 5$ Å, $k_z$ broadening is estimated to be $\Delta k_z \approx 1/\lambda \approx 0.2$ Å$^{-1}$,

~17% of the BZ), and the contribution of dispersion away from the Dirac node could contribute to a gap-like feature, considering the strong $k_z$ dispersion in this material. A more conclusive explanation requires further studies. The type-II characteristic is also reflected in the constant energy contours (Figs. 3n, o). The three dimensional intensity map E-$k_x$-$k_z$ shows an electron pocket (*red arrow*) and a hole pocket (*blue arrow*) approaching each other near the Dirac point energy, which is another important signature of type-II Dirac cones. This anisotropic touching between the electron and hole pockets contributes finite density of states around the Dirac point and this is distinguished from the vanishing density of states at the Dirac point in type-I Dirac semimetal.

**Theoretical calculation.** In order to further reveal the topological nature of surface states and type-II Dirac cone in PtTe$_2$, we present the first-principles calculations of electronic structure and perform symmetry analysis. Figure 4a shows the calculated band structure along both the in-plane S-D-T and out-of-plane A-D-Γ directions through the D point. Due to the band inversion between $\Gamma_4^+$ and $\Gamma_4^-$ at the Γ point, there is a topologically

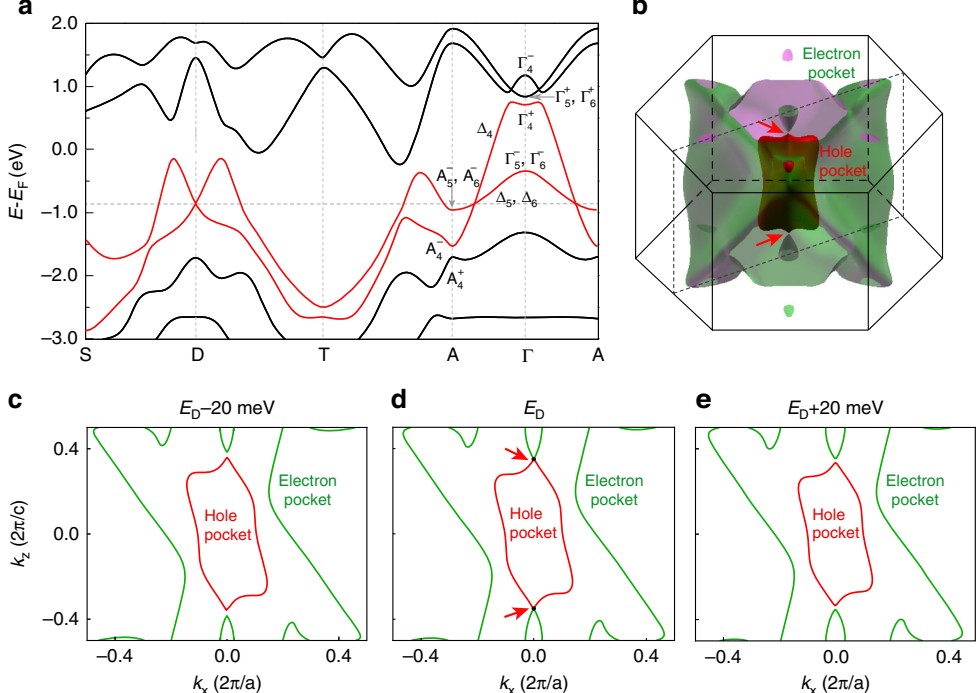

**Fig. 4** Theoretical calculation of type-II Dirac cone in PtTe$_2$. **a** Calculated band dispersion along the in-plane direction S-D-T and out-of-plane direction A-D-Γ-D-A through the Dirac point. **b** Three-dimensional plot of the electron and hole pockets at the Dirac point energy. The electron and hole pockets touch at the Dirac point D. **c-e** Contours of electron and hole pockets at −20, 0, + 20 meV around the Dirac point energy in the $k_x$–$k_z$ plane. The electron and hole pockets touch at $E_D$ and clearly separated at $E_D \pm 20$ meV

nontrivial gap between them which gives rise to the surface states connecting the gapped cone structure at the M′ point. In addition, there is another band inversion at the A point at $\sim -2$ eV, which leads to the existence of the deep surface state as mentioned above. We have calculated the $Z_2$ invariants for bands below these two gaps, confirming the nontrivial topology of them. The bulk Dirac cone is formed by two valence bands with Te-$p$ orbitals (highlighted by *red* color). These two bands show linear dispersions in the vicinity of D along both the in-plane (S-D-T) and out-of-plane (Γ-A) directions, confirming that it is a three-dimensional Dirac cone. This band crossing is unavoidable, because these two bands belong to different representations ($\Delta_4$ and $\Delta_{5+6}$) respectively, as determined by the $C_3$ rotational symmetry about the $c$ axis[2]. The different irreducible representations prohibit hybridization between them, resulting in the symmetry-protected band crossing at $D = (0,0,0.346c^*)$. As each band is doubly degenerate, the band crossing forms the four-fold degenerate Dirac point. We also calculated the energy contours by tuning the chemical potential to the Dirac point, as shown in Fig. 4b. It is clear that there is a hole pocket in the BZ center (*red* color), while the much more complicated electron pockets (*green* color) are composed of a large outer pocket and a small inner one. The hole pocket and the small electron pocket touch each other at two Dirac points as shown in the isoenergy counter in the $k_x$–$k_z$ plane (Fig. 4d). By tuning the chemical potential above or below $E_D$ (Figs. 4c, e), we find that the hole and electron pockets disconnect, confirming that they only touch at the Dirac point.

## Discussion

By combining both ARPES measurements and first-principles calculations, we provide the first direct experimental evidence for the realization of type-II 3D Dirac fermions in single crystal PtTe$_2$. While there are trivial bands at the Dirac nodes of type-II

semimetals, their magneto-transport response is different from Dirac fermions. As long as the topologically nontrivial bands cross the Fermi level, they can contribute to distinctive transport behavior even in the presence of trivial bands[33, 34]. Recent transport measurements have revealed a nontrivial Berry phase in a sister compound PdTe$_2$ with the Dirac node also below $E_F$[35], and similar transport properties can be expected in PtTe$_2$ if the Fermi level is tuned so that both bands forming the Dirac cone cross the Fermi level. By further tuning the Dirac node to the Fermi energy, more intriguing transport properties (e.g., angle-dependent negative magnetoresistance) can be further revealed. Doping PtTe$_2$ by Ir is one possible solution. IrTe$_2$ has the same crystal structure as PtTe$_2$ at room temperature, yet with the Dirac node above $E_F$ (Supplementary Fig. 2). By substituting Ir for Pt in Pt$_{1-x}$Ir$_x$Te$_2$, it is possible to fine tune the Dirac point to the Fermi energy. Expanding the $c$ axis lattice constant either by tensile strain[27] or appropriate intercalation[36] can also shift the Dirac node toward the Fermi level, and finding a suitable intercalant is critical along this line. Our study paves the way for designing and realizing a number of similar type-II Dirac materials in same chemical group ($PCh_2$, $P = $ Pt,Pd; $Ch = $ Se,Te)[27]. Moreover, the realization of type-II Dirac semimetal provides a new platform for investigating various intriguing properties different from their type-I analogues such as anisotropic magneto-transport properties.

## Methods

**Sample growth**. High quality PtTe$_2$ single crystal was obtained by self-flux methods. High purity Pt granules (99.9%, Alfa Aesar) and Te lump (99.9999%, Alfa Aesar) at a molar ratio of 2:98, were loaded in a silica tube, which is flame-sealed in a vacuum of ∼ 1 Pa. The sample was heated at 700 °C for 48 hours to homogenize the starting materials. The reaction was then slowly cooled to 480 °C at 5 °C h$^{-1}$ to crystallize PtTe$_2$ in Te flux. The excess Te was centrifuged isothermally after 2 days.

**ARPES measurement**. ARPES measurements were taken at BL13U of Hefei National Synchrotron Radiation Laboratory, BL9A of Hiroshima Synchrotron

Radiation Center under the proposal No.15-A-26 and our home laboratory. The crystals were cleaved in situ and measured at a temperature of T ≈ 20 K in vacuum with a base pressure better than $1 \times 10^{-10}$ torr.

**Theoretical calculation**. All first-principles calculations are carried out within the framework of density-functional theory using the Perdew-Burke-Ernzerhof-type[37] generalized gradient approximation for the exchange-correlation potential, which is implemented in the Vienna ab initio simulation package[38]. A $8 \times 8 \times 6$ grid of **k** points and a default plane-wave energy cutoff (230 eV in this case) are adopted for the self-consistent field calculations. We use the Methfessel-Paxton-type smearing method with a width of 0.2 eV. Spin-orbit coupling is taken into account in our calculations. We calculate the surface spectral function and Fermi surface using the surface Green's function method[39] based on maximally localized Wannier functions[40] from first-principles calculations of bulk materials.

**Data availability**. The data that support the plots within this paper and other findings of this study are available from the corresponding author upon reasonable request.

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

## Acknowledgements

This work is supported by the National Natural Science Foundation of China (Grant No. 11334006, 11427903 and 11674188), Ministry of Science and Technology of China (Grant No. 2015CB921001, 2016YFA0301001 and 2016YFA0301004) and Beijing Advanced Innovation Center for Future Chip (ICFC).

## Author contributions

S.Z. designed the project. M.Y., K.Z., E.Y., W.Y., K.D., G.W., H.Z. and S.Z. performed the ARPES measurements and analyzed the ARPES data. H.H., W.D. performed theoretical calculation, K.Z., Y.W. prepared the single crystals. H.Y. discussed the data. M.Y. and S.Z. wrote the manuscript, and all authors commented on the manuscript.

## Additional information

**Competing interests:** The authors declare no competing financial interests.

