## [Peer Review File · Nature Communications]

Reviewers' comments:

Reviewer #1 (Remarks to the Author):

Yan et al. report evidence of type-II Dirac fermions in PtTe₂ using a combination of ab initio calculations and ARPES experiments. The field of Dirac/Weyl semimetals has received intensive interest in recent years, and as mentioned in the paper, there are four distinct classes of such quasiparticles that can in principle exist: type-I and type-II Dirac fermions and type-I and type-II Weyl fermions. So far, all but type-II Dirac fermions have been published in the literature. However, over the past few months, a series of theoretical and experimental works (including this paper) have been posted online reporting evidence of type-II Dirac fermions. Given that type-I Dirac and type-I and II Weyl fermions have all been high profile discoveries, I think the realization of a type-II Dirac fermion is a topic well deserving of publication in Nature Communications.

In the work by Yan et al., the band structure of PtTe₂ measured using ARPES agrees well with the ab initio calculations, which supports type-II Dirac fermions. The only unexpected feature they observe is a small apparent gap at the Dirac point (Fig. 3m) for reasons presently unknown. It would be helpful if the authors address whether this can simply be an artifact of a slight misalignment in k-space. Otherwise I find the work technically sound and recommend publication.

Reviewer #2 (Remarks to the Author):

Experimental identification of type-II Weyl and Dirac fermions is an important subject in the realization of novel topological quantum phases in condensed matter physics. Since it was theoretically predicted that the type-II Weyl fermions, a novel type of quasi-particles, can arise from a new class of topological semimetals (type-II Weyl semimetals), experimentalists have made great efforts to search this type of materials, as well as type-II Dirac semimetals which can host the spin-degenerate counterpart of type-II Weyl fermions, i.e. type-II Dirac fermions. The manuscript by M. Yan et al. is one of the examples along this line. By carrying out photoemission spectroscopy (ARPES) measurements, combined with first-principle calculation and symmetry analysis, M. Yan et al determined the detailed electronic structure of PtTe₂. They observed three distinct topological features in this transition metal dichalcogenide compound, 1) a three-dimensional Dirac cone at 1 eV below Fermi level, which is identified as type-II Dirac cone of bulk bands, 2) a surface Dirac cone at -2.1 eV in a bulk band gap, and 3) a bulk band gap slightly below Fermi level at the Dirac point (M). The results reported in this manuscript are original, the experimental data is of high quality. The manuscript is also well-written for conveying the authors' messages to readers. Although I find these results are interesting, in my opinion, the manuscript by M. Yan et al does not provide clear evidence that type-II Dirac fermions can or potentially could emerge in PtTe₂. The considerations are,

- 1) As quasi-particles, type-II Dirac fermions arise as low-energy excitations in topological semimetals. However, in PtTe₂ the three-dimensional type-II Dirac cone locates at about 1 eV below Fermi level. The binding energy is too high for exciting the states near the Dirac cone to Fermi level, and it is also difficult for those states contributing to transport properties.
- 2) In a metallic system it is difficult to shift chemical potential. From the determined electronic structure it seems that the density of states between the type-II Dirac cone point and the Fermi level is high. Do the authors have any consideration and/or estimation in mind about how to shift the Fermi level downwards, such that type-II Dirac fermions would emerge in PtTe₂? For instance, if by doping, how many holes need to be brought into a unit cell and is it achievable, and if by element substitution, what would be changes in the electronic structure?
- 3) A typo: the text "which is labeled as "D" in Fig. 1d." in the second last line of page 6 should be replaced with "which is labeled as "D" in Fig. 1e."

In conclusion, if the authors provide satisfactory answers to the questions/comments listed above, this manuscript deserves further consideration.

Reviewer #1 (Remarks to the Author):

Yan et al. report evidence of type-II Dirac fermions in PtTe₂ using a combination of ab initio calculations and ARPES experiments. The field of Dirac/Weyl semimetals has received intensive interest in recent years, and as mentioned in the paper, there are four distinct classes of such quasiparticles that can in principle exist: type-I and type-II Dirac fermions and type-I and type-II Weyl fermions. So far, all but type-II Dirac fermions have been published in the literature. However, over the past few months, a series of theoretical and experimental works (including this paper) have been posted online reporting evidence of type-II Dirac fermions. Given that type-I Dirac and type-I and II Weyl fermions have all been high profile discoveries, I think the realization of a type-II Dirac fermion is a topic well deserving of publication in Nature Communications.

Reply: We thank the reviewer for appreciating the scientific merits of our work and recommendation for publication. Our work is the first to report the experimental realization of type-II Dirac fermions and has motivated many research works to investigate the interesting properties of type-II Dirac semimetals.

In the work by Yan et al., the band structure of PtTe₂ measured using ARPES agrees well with the ab initio calculations, which supports type-II Dirac fermions. The only unexpected feature they observe is a small apparent gap at the Dirac point (Fig. 3m) for reasons presently unknown. It would be helpful if the authors address whether this can simply be an artifact of a slight misalignment in k-space. Otherwise I find the work technically sound and recommend publication.

Reply: Regarding the small gap at the Dirac point, we have repeated the experiment and scrutinized the k_z dispersion by taking a fine step (0.3 eV photon energy/step, corresponding to 0.012×2π/c in k_z, see Fig.R1 below) to exclude any misalignment in the k-space.

Fig.R1: Dispersions along Γ -M high symmetry direction at different k_z values from 0.28 to 0.41×2π/c with a fine photon energy step of 0.3 eV.

A more likely possibility is the k_z broadening due to the finite lifetime of photoelectrons. Namely, the measured dispersion is the averaged dispersion over a finite k_z window, and the contribution of dispersion away from the Dirac node could contribute to an apparent gap opening. At 22 eV photon energy, the escape length for photoelectrons is $\sim 5 \text{ \AA}$ according to the universal curve (C. J. Powell, *J. Electron Spectrosc. Relat. Phenom.* **47**, 197 (1988)). From the uncertainty principle, the k_z broadening is estimated to be $\Delta k_z \approx \lambda^{-1} \approx 0.2 \text{ \AA}^{-1}$, which is approximately 17% of the Brillouin zone ($2\pi/c$). The averaged dispersion over such Δk_z interval may contribute to a gap-like feature at the Dirac node, considering the strong k_z dispersion in this material.

Reviewer #2 (Remarks to the Author):

Experimental identification of type-II Weyl and Dirac fermions is an important subject in the realization of novel topological quantum phases in condensed matter physics. Since it was theoretically predicted that the type-II Weyl fermions, a novel type of quasi-particles, can arise from a new class of topological semimetals (type-II Weyl semimetals), experimentalists have made great efforts to search this type of materials, as well as type-II Dirac semimetals which can host the spin-degenerate counterpart of type-II Weyl fermions, i.e. type-II Dirac fermions. The manuscript by M. Yan et al. is one of the examples along this line. By carrying out photoemission spectroscopy (ARPES) measurements, combined with first-principle calculation and symmetry analysis, M. Yan et al determined the detailed electronic structure of PtTe₂. They observed three distinct topological features in this transition metal dichalcogenide compound, 1) a three-dimensional Dirac cone at 1 eV below Fermi level, which is identified as type-II Dirac cone of bulk bands, 2) a surface Dirac cone at -2.1 eV in a bulk band gap, and 3) a bulk band gap slightly below Fermi level at the Dirac point (M). The results reported in this manuscript are original, the experimental data is of high quality. The manuscript is also well-written for conveying the authors' messages to readers.

Reply: We thank the reviewer for appreciating the originality and high quality of our manuscript.

Although I find these results are interesting, in my opinion, the manuscript by M. Yan et al does not provide clear evidence that type-II Dirac fermions can or potentially could emerge in PtTe₂. The considerations are,

1) As quasi-particles, type-II Dirac fermions arise as low-energy excitations in topological semimetals. However, in PtTe₂ the three-dimensional type-II Dirac cone locates at about 1 eV below Fermi level. The binding energy is too high for exciting the states near the Dirac cone to Fermi level, and it is also difficult for those states contributing to transport properties.

Reply: We agree with the referee that tuning the Dirac point to the Fermi level will be important for certain transport properties, e.g. negative magnetoresistance. The topological property of the material, however, does not change with the energy position of the Dirac node. As long as the topological nontrivial bands cross the Fermi level, they can contribute to the transport behavior even in the presence of trivial bands (Z. Yu et

al., Phys. Rev. Lett. 117, 077202 (2016)). In fact, our work is the first experimental realization of type-II Dirac semimetal and has motivated many research groups to investigate the intriguing properties of type-II Dirac semimetals. A recent followed-up experimental work (F. Fei *et al.*, arXiv: 1611.08112) has reported non-trivial Berry phase in a similar material PdTe₂, which also occurs in the same crystal structure, with Dirac point energy at 0.53 eV below the Fermi level. We believe that with more theoretical and experimental efforts on this new class of materials, the Dirac point energy could be further tuned toward the Fermi energy, and more intriguing transport properties will continue to be discovered.

2) In a metallic system it is difficult to shift chemical potential. From the determined electronic structure, it seems that the density of states between the type-II Dirac cone point and the Fermi level is high. Do the authors have any consideration and/or estimation in mind about how to shift the Fermi level downwards, such that type-II Dirac fermions would emerge in PtTe₂? For instance, if by doping, how many holes need to be brought into a unit cell and is it achievable, and if by element substitution, what would be changes in the electronic structure?

Reply: Hole doping can in principle shift the Dirac node toward the Fermi energy. Our calculation shows that, a hole concentration of $6.45 \times 10^{21} \text{ cm}^{-3}$ can shift the Dirac node to Fermi level. Such doping level corresponds to 0.5 hole per unit cell (Fig. R2).

Fig. R2: (a) Calculated density of states near the Fermi level in pristine PtTe₂ single crystal. (b) Calculated band dispersion of PtTe₂ along the in-plane directions S-D-T and out-of-plane direction A-Γ-A after introducing 0.5 hole per unit cell.

We provide two possible methods to tune the Dirac node closer to the Fermi energy, and relevant discussions have been included in the revised manuscript.

a. *Chemical substitution/doping.* Element substitution/chemical doping has been widely used to tune the chemical potential. To induce holes in PtTe₂, substitutions of Ir/Rh

for Pt and As/Sb for Te are two possible solutions. In particular, preliminary calculations show that IrTe₂, which has the same crystal structure as PtTe₂ at room temperature, has a tilted Dirac node at 280 meV *above* the Fermi level, while the Dirac node in Pt_{0.5}Ir_{0.5}Te₂ is 390 meV *below* the Fermi level (Pt_{0.5}Ir_{0.5}Te₂). By fine-tuning the composition x, it is possible to tune the Dirac node energy of Pt_{1-x}Ir_xTe₂ to the Fermi energy. Such Pt_{1-x}Ir_xTe₂ system is interesting for further investigation.

Fig. R3: (a) Calculated band dispersion of IrTe₂ along high symmetry directions, resolving the tilted Dirac node above the Fermi level. (b) Calculated band dispersion of Ir_{0.5}Pt_{0.5}Te₂ along high symmetry directions.

- b. *Uniaxial strain or intercalation.* According to the calculation in PtSe₂ (H. Huang *et al.*, *Phys. Rev. B* **94**, 121117(R) (2016), supplementary information), a tensile strain along the c axis could effectively shift the Dirac node by ~900 meV towards the Fermi energy without changing its topological properties. Similarly, when a 10% tensile strain is applied along the c axis, the Dirac node in PtTe₂ could approach the Fermi level (Fig. R4). Experimentally, an external strain or intercalation could have equivalent effect to increased layer separation. Similar intercalations have been successfully realized in (Li_{0.8}Fe_{0.2})OH intercalated FeSe layers (X. Lu *et al.*, *Nature Mater.* **14**, 325-329 (2015)) which increases the c axis lattice constant effectively (by 68%), and Pd intercalated IrTe₂ (J. J. Yang *et al.*, *Phys. Rev. Lett.* **108**, 116402 (2012)) albeit with a much smaller change. Finding a suitable intercalant is critical along this line.

Fig. R4: Calculated band dispersion of PtTe₂ along Γ -A direction after applying 5% and 10% tensile strain along c axis, respectively.

To summarize, tuning the Dirac node to the Fermi level is technically feasible. Further transport experiment can be an independent research topic in the near future, but this does not affect the completeness and importance of our work on revealing the type-II Dirac cone for the first time by high-resolution ARPES experiment.

3) A typo: the text “which is labeled as “D” in Fig. 1d.” in the second last line of page 6 should be replaced with “which is labeled as “D” in Fig. 1e.”

Reply: We thank the reviewer for pointing it out and we have corrected it in the revised manuscript.

In conclusion, if the authors provide satisfactory answers to the questions/comments listed above, this manuscript deserves further consideration.

Reviewers' comments:

Reviewer #1 (Remarks to the Author):

I am satisfied with the additional fine k_z data that the authors have presented in Fig.S1 to rule out misalignment. I recommend publication.

Reviewer #2 (Remarks to the Author):

I thank the authors made detailed reply to my previous comments, which partially answered the questions raised in my first report. However, concerning the Dirac fermions, I am not sure that they addressed my main point. In the reply to my first question, they wrote "The topological property of the materials, however, does not change with the energy position of the Dirac node. ...". Actually, this an important issue which should clearly discussed in the manuscript. Here is the consideration: a topological 3D Dirac semimetal can be viewed as a composite of two sets of Weyl fermions with opposite chiralities (+1 and -1). In a Weyl semimetal all the bands which form a Weyl node should cross Fermi level, which results in a non-zero Chern number when integrating the Berry curvature over the Fermi surface. In an analogy, all the bands which form a Dirac node should also cross Fermi level. From Fig. 4a in the manuscript, however, it seems to me that only one band with δ_4 symmetry along the k_z axis crosses the Fermi level.

In conclusion, if the authors can provide satisfactory answer(s) to the point mentioned above, I will recommend this work for publication in Nature Communications.

A note: I think the occupied electronic structure in PdTe₂ (arXiv: 1611.08112) is qualitatively different to PtTe₂ (this manuscript), i.e. in PdTe₂ both bands which form the Dirac cone cross Fermi level along k_z axis.

Reviewer #1

I am satisfied with the additional fine k_z data that the authors have presented in Fig.S1 to rule out misalignment. I recommend publication.

Reply: We thank the reviewer for recommendation for publication.

Reviewer #2

I thank the authors made detailed reply to my previous comments, which partially answered the questions raised in my first report. However, concerning the Dirac fermions, I am not sure that they addressed my main point. In the reply to my first question, they wrote “The topological property of the materials, however, does not change with the energy position of the Dirac node. ...”. Actually, this an important issue which should clearly discussed in the manuscript. Here is the consideration: a topological 3D Dirac semimetal can be viewed as a composite of two sets of Weyl fermions with opposite chiralities (+1 and -1). In a Weyl semimetal all the bands which form a Weyl node should cross Fermi level, which results in a non-zero Chern number when integrating the Berry curvature over the Fermi surface. In an analogy, all the bands which form a Dirac node should also cross Fermi level. From Fig. 4a in the manuscript, however, it seems to me that only one band with δ_4 symmetry along the k_z axis crosses the Fermi level.

In conclusion, if the authors can provide satisfactory answer(s) to the point mentioned above, I will recommend this work for publication in Nature Communications.

A note: I think the occupied electronic structure in PdTe₂ (arXiv: 1611.08112) is qualitatively different to PtTe₂ (this manuscript), i.e. in PdTe₂ both bands which form the Dirac cone cross Fermi level along k_z axis.

Reply: We agree that a topological 3D Dirac point can be viewed as a pair of Weyl points with opposite chirality. In a Weyl semimetal, the Weyl point can serve as the magnetic monopole of in momentum space where the Berry curvature acts as an effective magnetic field. Similar to the Gauss’s law in electrodynamics, the topological nature of Weyl points can be captured by the topological charge, which is calculated by integrating the Berry curvature of N bands in an arbitrary closed surface enclosing the Weyl point [Soluyanov *et al. Nature* 527, 495 (2015)]. The N bands have an energy gap to the higher bands in the closed surface but touch higher bands at the Weyl point. The topological charge would not change as long as the Weyl point is not destroyed. To be more specific, the topological property of the Dirac point does not change with the energy position of the Dirac node.

Some measurable physical quantities such as magneto-transport responses are dominated by the states around the Fermi energy. These properties are, of course, affected by the position of Fermi level. However, as mentioned in the previous reply, the Fermi level of PtTe₂ can, in principle, be tuned to the Dirac point by chemical doping, uniaxial strain or intercalation so that both bands forming the Dirac cone cross the

Fermi level similar to the case of PdTe₂. Hence, the topological properties can be observable in PtTe₂ with proper Fermi level tuning, which is an interesting aspect for further research. Indeed, we are glad to see that our work has motivated research work to investigate the intriguing properties of type-II Dirac semimetals, including the PdTe₂ work pointed out by the referee.